# Personality Traits, Burnout, and Psychopathology in Healthcare Professionals in Intensive Care Units—A Moderated Analysis

**DOI:** 10.3390/healthcare12050587

**Published:** 2024-03-04

**Authors:** Varvara Pakou, Dimitrios Tsartsalis, Georgios Papathanakos, Elena Dragioti, Mary Gouva, Vasilios Koulouras

**Affiliations:** 1Intensive Care Unit, University Hospital of Ioannina, University of Ioannina, 45500 Ioannina, Greece; varvarapak@uoi.gr (V.P.); gppthan@icloud.com (G.P.); vpkoulouras@yahoo.gr (V.K.); 2Laboratory of Psychology of Patients, Families & Health Professionals, Department of Nursing, School of Health Sciences, University of Ioannina, 45500 Ioannina, Greece; dragioti@uoi.gr (E.D.); gouva@uoi.gr (M.G.); 3Department of Clinical Physiology, Sundsvall Hospital, 85643 Sundsvall, Sweden

**Keywords:** personality, burnout, nurses, intensivists, intensive care unit, psychopathology

## Abstract

This study explored the associations between personality dimensions, burnout, and psychopathology in healthcare professionals in intensive care units (ICUs). This study further aimed to discern the differences in these relationships when considering the variables of critical care experience (less than 5 years, 5–10 years, and more than 10 years), profession (nurses versus intensivists), and the urban size of the city where the ICU is located (metropolitan cities versus smaller urban cities). This cross-sectional investigation’s outcomes are based on data from 503 ICU personnel, including 155 intensivists and 348 nurses, in 31 ICU departments in Greece. Participants underwent a comprehensive assessment involving a sociodemographic questionnaire, the Eysenck Personality Questionnaire (EPQ), the Maslach Burnout Inventory (MBI), and the Symptom Checklist-90 (SCL-90). To analyze the interplay among critical care experience, burnout status, and psychopathology, a moderation analysis was conducted with personality dimensions (i.e., psychoticism, extraversion, and neuroticism) serving as the mediator variable. Profession and the urban size of the ICU location were considered as moderators influencing these relationships. Male healthcare professionals showed higher psychoticism levels than females, aligning with prior research. Experienced nurses reported lower personal achievement, hinting at potential motivation challenges for professional growth. Psychoticism predicted high depersonalization and low personal achievement. Neuroticism and psychoticism negatively impacted ICU personnel’s mental well-being, reflected in elevated psychopathology scores and burnout status. Psychoticism appears to be the primary factor influencing burnout among the three personality dimensions, particularly affecting intensivists. In contrast, nurses are more influenced by their critical care experience on their mental health status.

## 1. Introduction

The term “burnout” was coined by Freudenberger in 1974, who observed declining motivation and commitment among mental health clinic volunteers [1]. Maslach further advanced the concept, creating the widely used Maslach Burnout Inventory (MBI) [2]. According to this framework, burnout syndrome involves emotional exhaustion, physical fatigue, and cognitive weariness due to prolonged uncontrolled work-related stressors [3]. The syndrome comprises three main dimensions: high emotional exhaustion, high depersonalization, and low personal achievement [4]. High emotional exhaustion involves work-related fatigue, marked by feelings of energy depletion, whereas high depersonalization is a defense mechanism, leading to distancing from work through negativism or cynicism. Low personal achievement reflects frustration with work-related achievements, resulting in reduced professional efficiency [4].

Burnout is influenced by enduring personal and environmental stressors, such as major illness, family difficulties, or persistent adversity [5]. In healthcare, burnout syndrome and stress-related disorders are prevalent, impacting various roles like nurses [6,7], emergency room physicians [8], and intensive care unit (ICU) physicians [9,10]. Mental health professionals are also susceptible to high levels of burnout syndrome [11] due to the significant emotional and cognitive stress associated with their profession [12]. Factors associated with physician burnout include age, sex, marital status, personality traits, ICU work experience, work environment, workload, shift work, ethical issues, and end-of-life decision-making [13].

Notably, burnout in physicians is associated with increased risks of major medical errors, reduced patient care quality [14,15], absenteeism, decreased productivity [16,17,18], and impaired team relationships [19], leading to worsened care quality and higher healthcare costs [3,20]. Nurses often experience burnout due to factors like high workload, value incongruence, low job control, limited decision latitude, poor social climate/support, and insufficient rewards [21]. A common factor across all professions may be the problem of aggression faced by healthcare workers. This aggression, whether it comes from patients or workplace interactions, is considered a precursor to burnout [22,23,24].

Burnout has emerged as a pervasive issue among healthcare professionals [3]. For example, Woo et al. [25] found that 11.23% of nurses globally have experienced symptoms of burnout. In the field of emergency medicine, healthcare workers exhibit a burnout prevalence rate of 20–60% [26]. Similarly, 40% of mental health professionals have encountered burnout [11], and Rotenstein et al. [27] report that 67% of physicians have demonstrated overall burnout prevalence estimates. In the ICU setting, burnout is reported more prevalently among nurses than physicians [28,29], with direct implications for work, personal functionality, and overall healthcare services [30].

A challenge in current burnout research is the absence of agreed-upon terminology, as demonstrated by the numerous definitions identified, such as in the comprehensive review by Rotenstein et al. [27], covering 182 studies and 109,628 individuals across 45 countries. This lack of consensus leads to widely varying reported prevalence rates, with substantial fluctuations in burnout rates among physicians and within specific subscales [27]. The prevalence of burnout in the ICU setting also shows significant variability [13,31].

Personality traits significantly impact burnout experiences under similar work conditions [32]. Utilizing the dimensions of psychoticism (P), extraversion (E), and neuroticism (N), also known as Eysenck’s PEN model [33,34], this study defines personality through these three dimensions [35]. The PEN model posits that these dimensions represent fundamental aspects of an individual’s personality, influencing their behavior, cognition, and emotional responses [33,34,36]. Specifically, psychoticism is associated with traits such as disagreeableness, non-conscientiousness, and a propensity for risk-taking. Extraversion encompasses characteristics like sociability, assertiveness, positive emotions, and impulsivity, while neuroticism involves traits such as anxiety, depression, and self-doubt. Furthermore, these personality dimensions—P, E, and N—are considered orthogonal [37], indicating no correlation between them, thus contributing to a comprehensive understanding of various facets of an individual’s personality [36].

Neuroticism and lower extraversion are commonly associated with elevated burnout levels among nurses and physicians [29,38,39,40,41,42,43]. However, there is a notable scarcity of studies comparing personality traits between nurses and physicians [44], particularly within the ICU environment. Understanding such distinctions could have far-reaching implications for team dynamics, patient care, workplace satisfaction, and healthcare system improvement.

Personality differences are observed among rural and urban residents, with environmental factors influencing personality structure and exacerbating feelings of anxiety and depression [45,46]. Urban characteristics, such as exposure to heavy metals, pesticides, bisphenol A (BPA), noise pollution, and poor air quality, are linked to mental health conditions [47,48,49,50,51,52]. In Greek metropolitan areas like Athens and Thessaloniki, high population density and limited green spaces pose challenges [53]. Despite these, specialized hospitals in these regions serve wider areas in southern and northern Greece. The lower nurse-to-ICU bed ratio in metropolitan hospitals during COVID-19 [54] suggests an added psychological burden on ICU personnel, raising questions about the mental impact on healthcare workers in large urban areas compared to smaller cities.

### Aim of the Study

This study delves into the intricate connections among socio-demographics, personality traits, psychopathology, and burnout in ICU personnel. Using a moderated mediation model, it explores the nuanced interplay involving critical care experience, personality dimensions, burnout, profession, and hospital location. Critical care experience serves as the independent variable, with personality dimensions mediating the relationship between experience and burnout. Additionally, the nurse or intensivist profession and the urban status of the hospital act as moderators, examining variations based on professional role and urban size. This design allows for assessing both direct and mediated effects as well as investigating whether the mediation process varies with moderator variables. In essence, we tested the moderated mediation model illustrated in Figure 1.

Among other inquiries, the model addresses the following research questions:How do personality dimensions differentiate as age increase?How do personality traits vary between male and female ICU personnel?What is the relationship between critical care experience, burnout, and psychopathology?Does the relationship between critical care experience, burnout, and psychopathology vary depending on the profession or the urban size of the ICU location?How do personality dimensions and profession interact in predicting burnout?Does the effect of critical care experience on burnout and psychopathology vary between nurses and intensivists or between ICU location in small and large cities?

Understanding these connections is vital for theoretical advancement and targeted interventions to improve the well-being of ICU personnel. This research has the potential to contribute to tailored support systems, addressing the unique needs of intensivists and nurses in high-stress environments.

## 2. Methods

### 2.1. Participants and Study Design

This study involved a cross-sectional investigation conducted with a sample of 573 respondents drawn from 31 ICUs in 10 cities across Greece. The sampling process took place between December 2021 and June 2022. Out of the total 573 ICU personnel, 23 individuals were excluded from subsequent analysis as they did not fit the roles of nurses or physicians. Additionally, from the remaining 550 participants, 47 were identified with at least one missing value in the required demographic or professional characteristics, leading to their exclusion from further analysis. Consequently, the results presented in this study are based on data obtained from the remaining 503 respondents, comprising 155 intensivists and 348 nurses.

### 2.2. Measurements

The participants were asked to anonymously complete a sociodemographic questionnaire, which encompassed necessary personal and professional information. Additionally, they were required to fill out the adapted to Greek-language versions of the Maslach Burnout Inventory (MBI) [55], the Eysenck Personality Questionnaire (EPQ) [56], and the Symptom Checklist 90 (SCL-90) [57].

The MBI, developed by Maslach and Jackson [2], assesses burnout through 22 items grouped into three subscales: emotional exhaustion, depersonalization, and personal achievement. Respondents rate each item on a 7-point Likert scale ranging from ‘never’ (0) to ‘daily’ (6). The scale has been translated into Greek and has shown satisfactory validity and reliability [55]. In this study, we used the established thresholds on the three subscales, based on the adapted Greek version by Zis et al. [58]. Specifically, burnout was identified for scores ≥ 34 in emotional exhaustion, ≥13 in depersonalization, and ≤29 in personal achievement.

The EPQ, developed by Eysenck and Eysenck [35], assesses personality traits using a 90-item questionnaire. It measures three primary dimensions: extraversion, neuroticism, and psychoticism. Respondents rate each item on a binary scale (yes/no response). The scale has been translated into Greek and has shown satisfactory validity and reliability [56].

The SCL-90, developed by Derogatis [59], assesses a wide range of psychological symptoms. It comprises 90 questions across 9 subscales, namely somatization, obsessive-compulsive, interpersonal sensitivity, depression, anxiety, hostility, phobic anxiety, paranoid ideation, and psychoticism. Participants rate items on a 5-point Likert scale ranging from 0 (not at all) to 4 (extremely) based on distress level. Additionally, three global indices (global severity index [GSI], positive symptom distress index [PSDI], and positive symptom total [PST]) summarize psychological distress. The scale has been translated into Greek and validated for criterion and convergent validity [57]. This study reports data only for the GSI index, a measure that reflects the overall psychopathology. A higher GSI score indicates higher levels of psychological distress.

Participants provided verbal informed consent prior to their involvement in the study, and ethical approval was granted by the Ethics Committee of the University Hospital of Ioannina (protocol code 3263, dated 1 February 2019). Participants were also assigned unique identifiers to anonymize their data, and only authorized researchers had access to the raw data.

### 2.3. Statistical Analysis

Initial data analysis involved summarizing measures of central tendency and dispersion, with means and standard deviations reported for continuous variables and proportions for categorical ones. To compare groups, we employed independent samples *t*-tests, chi-squared tests of independence, and one-way ANOVA. To evaluate the internal consistency and reliability of the utilized scales—namely, the MBI, the EPQ, and the SCL-90—Cronbach’s alpha analysis was conducted [60]. Additionally, a Pearson correlation analysis was executed to examine the correlation matrix, encompassing age, personality traits, burnout, and psychopathology.

To evaluate the theoretical model, a moderated mediation approach was employed, wherein the relationship between critical care experience and burnout was mediated by three personality dimensions, while profession moderated this relationship (Figure 1). Model 76 of the PROCESS function for R, developed by Hayes [61], was utilized four times: once for each facet of burnout (i.e., emotional exhaustion, depersonalization, and personal achievement) [58] and once for the psychopathology, as indicated by the GSI [57]. This model was chosen for its robust capabilities in handling moderated mediation models [61].

Furthermore, we categorized critical care experience into three groups: less than 5 years, 5–10 years, and more than 10 years. Each group was compared to the preceding group using a sequential coding scheme [61]. This approach facilitated the comparison of successive levels of experience and enabled a nuanced exploration of the progressive nature of phenomena within the ICU setting. In both the moderation analysis of personality dimensions and the moderation analysis of burnout and psychopathology, main effects and interaction effects were examined. Bootstrap analysis was conducted to evaluate the indirect effects of the theoretical model. Unstandardized coefficients (B) and 95% confidence intervals were reported. Additionally, we reported the R^2^ for each model to assess the proportion of variance explained. Significant interaction effects were graphically illustrated. The statistical analysis was conducted under the guidance of a statistician to ensure accuracy and rigor in the interpretation of results.

A two-sided alpha level of 0.05 was set for determining statistical significance. All data were analyzed using the R (version 4.2.3) statistical language [62] equipped with the PROCESS function [61].

## 3. Results

### 3.1. Sample Demographics

Among the 503 respondents, nurses were notably younger (39.6 ± 7.8 vs. 45.7 ± 8.1, t (501) = 7.935, *p* < 0.001), more likely to be female (74.4% vs. 51.6%, c2(1) = 25.397, *p* < 0.001), and had less experience in the ICU environment compared to the intensivists (c2(2) = 20.579, *p* < 0.001).

### 3.2. Personality, Burnout, and Psychopathology Profiles

The psychometric scales used in this study demonstrated acceptable internal reliability, as evidenced by the examination of Cronbach’s alpha (a) coefficient (Table 1) [60]. The three personality traits were weakly correlated between them, confirming the assumed orthogonality [36,37]. Additionally, a notable positive correlation was reported between emotional exhaustion and depersonalization.

Table 2 presents the descriptive statistics of personality traits, burnout scores, and psychopathology for the total sample as well as stratified by profession, sex, critical care experience, and urban size of ICU location.

As shown in Table 3, a total of 129 participants, 38 intensivists and 91 nurses, were classified as experiencing high emotional exhaustion. Additionally, 195 participants, 43 intensivists and 152 nurses, were identified as having low personal achievement. Furthermore, 188 participants, 59 intensivists and 129 nurses, were categorized as experiencing high levels of depersonalization (Table 3).

The profession in the ICU was not related with the status of a high emotional exhaustion (c2(1) = 0.15, *p* = 0.699) and high depersonalization (c2(1) = 0.045, *p* = 0.831), while it was more probable for nurses to report a low personal achievement score (c2(1) = 11.473, *p* = 0.001). The GSI was significantly correlated with emotional exhaustion and depersonalization scores and exhibited a negative correlation with the personal achievement score. Specifically, the GSI was significantly higher among the respondents characterized as individuals experiencing high emotional exhaustion (1.37 vs. 0.72, t (501) = 9.508, *p* < 0.001) as well as among those characterized as individuals with high depersonalization (1.23 vs. 0.68, t (501) = 8.828, *p* < 0.001) or low personal achievement (1.05 vs. 0.79, t (501) = 4.098, *p* < 0.001).

### 3.3. Moderation Analysis of Personality Dimensions

The effect of demographics and the experience in the ICU environment on personality dimensions are presented in Table 4. Age was not found to have a significant effect on the three personality dimensions. Instead, a sex-based disparity was reported, with male healthcare professionals demonstrating markedly elevated levels of psychoticism compared to their female counterparts (M = 6.3 vs. M = 5.5, *p* < 0.001). A noteworthy main effect of profession on neuroticism emerged, indicating that nurses exhibited higher levels of neuroticism in comparison to intensivists (M = 11.1 vs. M = 9.9, *p* = 0.038).

Furthermore, our analysis showed some significant interaction effects (Table 2), which are further illustrated in Figure 2, Panel A–C. A noteworthy interaction between critical care experience and urban size of the ICU location concerning neuroticism was observed, suggesting that ICU personnel in minor cities tend to report higher levels of neuroticism across all experience levels than those in metropolitan cities, with the greatest difference seen in the group with more than 10 years of experience (Figure 2, Panel A). Additionally, the analysis revealed a significant interaction between critical care experience and profession regarding psychoticism and extraversion. Specifically, as experience in the ICU increases, nurses exhibited higher levels of psychoticism and lower levels of extraversion (Figure 2, Panel B,C).

### 3.4. Moderation Analysis of Burnout Facets and Psychopathology

The analysis uncovered notable associations between age and various dimensions of psychological well-being among the participants (Table 5). Specifically, a significant positive effect emerged between age and the likelihood of experiencing high emotional exhaustion (B = 0.037, *p* = 0.027), signifying a 1.4 times increase in the odds of high emotional exhaustion for each additional decade in the respondents’ age. Conversely, a negative relationship was identified between age and the likelihood of experiencing low personal accomplishment (B = −0.050, *p* = 0.004), suggesting a 0.6 times decrease in the odds of low personal accomplishment for each additional decade in the respondents’ age. Additionally, a noteworthy positive association was observed between age and the GSI (B = 0.009, *p* = 0.020), indicating a 0.09 increase in the GSI score for each additional decade in the respondents’ age. Furthermore, the personality trait of neuroticism (B = 0.082, *p* = 0.011) was identified as significant regressor of the GSI score, emphasizing its meaningful contribution to the overall psychological well-being of the participants (Table 5).

Additionally, the analysis revealed significant interaction effects (Table 5) between psychoticism, critical care experience, profession, and the urban size of the ICU location, which are further illustrated in Figure 3, Panel A–D. Notably, with an increasing duration of critical care experience, nurses tended to report lower personal achievement scores, whereas intensivists showed an opposite trend (Figure 3, Panel A). Moreover, the relationship between psychoticism and psychopathology, as measured by the GSI, varied in strength depending on the urban size of the ICU location. Specifically, ICU personnel in larger metropolitan areas exhibited a more robust relationship between psychoticism and GSI scores than those in smaller urban settings (Figure 3, Panel B). Lastly, there was a significant association between psychoticism and both low personal achievement and high depersonalization scores. This relationship was particularly strong among intensivists when compared to nurses (Figure 3, Panel C,D).

### 3.5. Indirect Effects

In the context of the examined model, the bootstrap analysis showed that personality dimensions did not have a significant indirect effect on the relationship between critical care experience and both psychopathology and burnout.

## 4. Discussion

### 4.1. Personality Dimensions

The three personality traits exhibited weak correlations (Pearson’s r < 0.3), affirming the assumed orthogonality between psychoticism, extraversion, and neuroticism [36,37,63,64]. The observation that male healthcare professionals manifested notably elevated levels of psychoticism in comparison to their female counterparts substantiates the hypothesized association with psychoticism. In contrast, the inconclusive sex differences reported for extraversion align with analogous evidence in the literature [65,66], irrespective of the specific measurement instrument employed [36,67].

The extant literature lacks consensus regarding whether nurses lean towards extroversion or introversion, with several reports suggesting tendencies in either direction [44]. Particularly among ICU nurses, their personality traits were characterized by higher scores on dominance, rebelliousness, and self-sufficiency, coupled with lower scores on emotional sensitivity and imagination compared to nurses from other departments [68,69], predominantly indicative of elevated psychoticism. The observation that, with increasing experience in the ICU, nurses exhibit higher levels of psychoticism and diminished levels of extraversion further elucidates the reported relations. This suggests that in the high-pressure environment of an ICU, nurses may develop heightened psychoticism and reduced extraversion as adaptive coping mechanisms in response to stress and the demanding nature of their roles [70].

This phenomenon may be linked to challenges faced by nurses in establishing effective communication with patients who often experience objective communication difficulties [71,72] and frequently undergo negative emotional reactions such as frustration, stress, anxiety, and depression [73,74]. Given that effective human-level interaction forms the cornerstone of a positive working environment for nurses, it is plausible to posit that persistent challenges in communication may predispose nurses to heightened feelings of anxiety, depression, self-doubt, and other negative emotions, contributing to an elevated psychoticism score. In this context, previous assertions highlighting inferior personality factors linked to communication skills in ICU nurses compared to hospitalization unit nurses [75] should not be attributed to systematic personality variance among nurses. Instead, it is crucial to recognize that these disparities may originate from substantial barriers to effective communication [76].

Contrastingly, as experience in the ICU increases, intensivists experience heightened neuroticism, indicating an increased sense of responsibility and concern for patient outcomes. However, this also implies a higher emotional and psychosomatic toll in managing critically ill patients over time in the ICU. Given that elevated neuroticism is typically associated with an increased risk of mental illness [77] and, on average, poorer outcomes in terms of health and relationship satisfaction [78,79], a necessity emerges for a more palliative approach for long-term ICU doctors.

### 4.2. Burnout and Psychopathology

No overall significant differences were observed between nurses and intensivists regarding burnout status, aligning with prior research findings [80,81]. However, this stands in contrast to other studies [28,29]. This discrepancy in findings might be attributable to the specific samples and settings of the various studies. In relation to personal achievement, an inverse relationship between age and the likelihood of a low score status was identified, implying that with increasing age and experience, healthcare professionals acquire enhanced coping mechanisms for the emotional and physical stressors inherent in ICU settings [82]. However, an intriguing dynamic is suggested by the discovery that intensivists with lower ICU experience exhibit a higher probability of reporting low personal achievement compared to ICU nurses with equivalent experience. This pattern is reversed among intensivists and nurses with greater ICU experience. The origin of this phenomenon remains uncertain but may be attributed to distinct professional obligations in an ICU setting, with intensivists typically assuming more leadership and decision-making responsibilities than nurses [83]. Consequently, lower-experience intensivists may find these responsibilities overwhelming early in their careers, resulting in a sense of lower personal achievement. Nevertheless, with accumulated experience, they may become more adept at managing complex cases, leading to an improved sense of accomplishment. Conversely, experienced nurses may perceive stagnation in their careers, with limited opportunities for advancement, autonomy, or professional growth. The absence of well-defined career paths can foster a sense of being trapped in their current position, potentially leading to increased job stress and a diminished sense of accomplishment [84,85,86].

Interestingly, a positive correlation was found between age and personal achievement among ICU personnel, with increasing age associated with a lower likelihood of reporting low personal accomplishment. This trend suggests that with advancing age, there is a discernible decrease in reporting low levels of personal accomplishment, indicating that older ICU personnel may perceive their professional achievements more positively. This can largely be attributed to the cumulative effect of accrued experience over time in relation to job satisfaction [87], which ostensibly contributes to a more favorable self-assessment of accomplishments in the challenging ICU environment. Therefore, the career trajectory of healthcare professionals in such settings might be characterized by a gradual and steady increase in personal accomplishment, underpinned by the development of adaptive skills and psychological resilience.

In contrast, there was a significant negative correlation between age and emotional exhaustion and psychopathology, confirming a general trend towards an increased feeling of emotional exhaustion in older ICU personnel, as reported in various studies [88,89,90]. This finding is another manifestation of the mental burden derived from the recent health crisis, resulting in a significant psychological toll and high burnout rates in the global healthcare community [81,91,92], particularly among Greek healthcare professionals [93,94,95,96]. Accordingly, the commonly reported notion that accumulated work experience and resulting physical and mental fatigue contribute to a higher overall psychopathology is confirmed [92,97,98,99].

The adverse implications of high neuroticism or high psychoticism scores are further evident in their direct effects on the psychopathology, aligning with prior reports [77,100,101]. Additionally, the added effect of stressors related to urban living, such as long commutes and a high cost of living, is reflected in the stronger relationship of personality traits like psychoticism with general psychopathology symptoms. This amplification could be particularly pronounced in high-stress healthcare settings, where the additional urban stressors can intensify the psychological impact of already demanding jobs. Studies have shown that environmental factors like urban living conditions can significantly affect mental health, potentially exacerbating underlying personality traits [102,103]. Additionally, the role of organizational culture in hospitals, especially in metropolitan areas, is critical. A less supportive work environment and limited mental health resources can leave ICU personnel, particularly those with higher psychoticism scores, more vulnerable to psychological distress and less equipped to cope effectively. The importance of organizational support in mitigating the impact of stress on mental health has been emphasized in various studies [104,105,106,107].

Secondly, the personality dimension of psychoticism was highlighted as an important predictor of the probability of high depersonalization and low personal achievement scores, confirming previous reports [38,108]. Specifically, feelings within the psychotic spectrum, such as irritability and quick-tempered reactions, a tendency to be suspicious of others’ motives or actions, feelings of restlessness, and general inner tension are valid signs of less psychological resilience and are significantly associated with a higher likelihood of perceiving incompetence, unsuccessful outcomes in their work, and a cynical behavior toward colleagues and patients [109]. Further, the association between psychoticism and lower personal achievement and higher depersonalization may be more pronounced among intensivists due to the increased responsibility and decisional burden associated with their roles, documented to impact decreased job satisfaction and emotional and psychological burnout [110,111]. Specifically, psychoticism, with its resilience traits, might influence how intensivists cope with the emotional challenges, potentially exacerbating the negative outcomes observed. Furthermore, the hierarchical nature of healthcare settings, combined with the intense and challenging work environment of the ICU, may amplify the association between psychoticism and adverse outcomes among intensivists. Nurses, although impacted, might not experience the same level of intensity and pressure in their roles.

Finally, this study found that personality traits do not significantly influence the relationship between critical care experience, psychopathology, and burnout. Essentially, the direct effects of critical care experience on both psychopathology and burnout are substantial enough to explain their relationships without the need to consider personality traits as an explanatory factor. This suggests that the impact of critical care experience on psychopathology and burnout occurs independently of an individual’s personality traits.

## 5. Conclusions

This study explored the relationships between personality traits, burnout, and psychopathology among ICU personnel, considering the moderating effect of experience and workplace setting. Among ICU nurses, increasing experience was correlated with higher psychoticism and lower extraversion, potentially as a coping mechanism for communication difficulties. Intensivists with lower ICU experience reported higher burnout. Psychoticism and neuroticism had direct effects on psychopathology, with stronger relations in metropolitan areas. Psychoticism predicted high depersonalization and low personal achievement, particularly among intensivists, indicating potential implications for coping with emotional challenges and adverse outcomes in high-pressure healthcare settings.

The findings underscore the need for a holistic understanding of the psychosocial dynamics within ICU settings, acknowledging the impact of communication challenges and the evolving responsibilities of healthcare roles. It is important to note that the repercussions of high levels of burnout and psychopathology among ICU personnel are significant, impacting both the healthcare professionals themselves [3,6,112] and the quality of patient care [113]. Research indicates that healthcare personnel experiencing burnout are associated with adverse patient outcomes. This includes diminished patient safety [114,115], increased standardized mortality ratios, and prolonged hospitalization durations [116]. Additionally, there is a correlation between healthcare worker burnout and reduced patient satisfaction [117] as well as an increased incidence of medical errors and malpractice [14,15,113].

Our study, therefore, lays a foundation for future explorations into strategic interventions aimed at establishing specialized training programs and robust support frameworks within the ICU. These programs and support systems are envisioned to offer both emotional and professional assistance to ICU personnel, thereby enhancing their ability to effectively manage the intense stress characteristic of the ICU and urban environment. Such proactive measures are crucial for maintaining the mental well-being and professional efficacy of ICU personnel.

### Limitations

It is imperative to acknowledge certain limitations in the interpretation and generalization of this study’s findings. Primarily, our results are based on a sample of ICU personnel from Greece, limiting generalizability beyond populations with similar cultural backgrounds, such as Mediterranean countries, cautioning against universal application to regions with distinct social structures. To enhance generalizability, future studies should include ICU personnel from various regions and cultural backgrounds. Conducting longitudinal research, as opposed to the cross-sectional approach used in this study, could offer insights into how these relationships evolve over time, particularly how personality traits interact with burnout and psychopathology in the long run. Additionally, comparing ICU personnel in urban versus rural settings, or in different healthcare systems, might reveal how environmental and systemic factors influence these relationships.

Secondly, the moderated mediation model employed in this research focused on the relationship between critical care experience, personality traits, burnout, and psychopathology. As such, the results are constrained by the specific pathways and variables within this model. While our study provides valuable insights into these particular associations, it does not capture the entirety of the complex interactions between personality dimensions and psychopathology. Other unexplored variables and pathways may exist, and the extent to which our findings can be extrapolated to a broader context may be limited. Exploring factors such as workplace environment, support systems, and personal life stressors could provide a more comprehensive understanding of the dynamics at play. Therefore, future research should consider additional factors and pathways to gain a more comprehensive understanding of the complex relationship between personality traits, burnout, and psychopathology in the context of ICU personnel. Finally, qualitative studies, such as interviews or focus groups, could provide deeper insights into the personal experiences and perspectives of ICU personnel, which might be missed in quantitative models.

## Figures and Tables

**Figure 1 healthcare-12-00587-f001:**
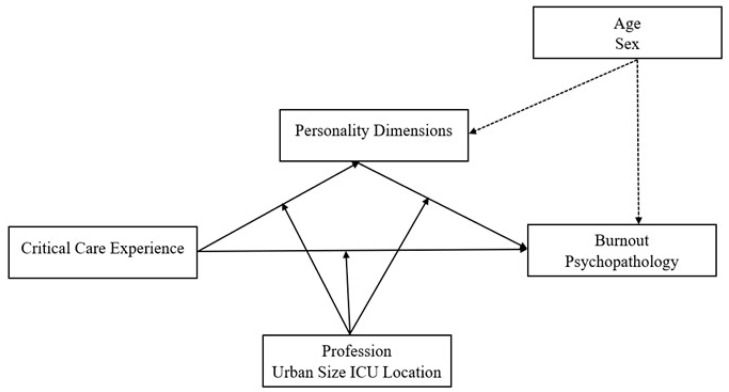
Theoretical moderated mediation model. Notes: Personality dimensions are hypothesized to mediate the relationship between critical care experience and burnout/psychopathology. In contrast, profession, and urban size of ICU location (and potentially age and sex) are hypothesized to moderate the relationships within the model.

**Figure 2 healthcare-12-00587-f002:**
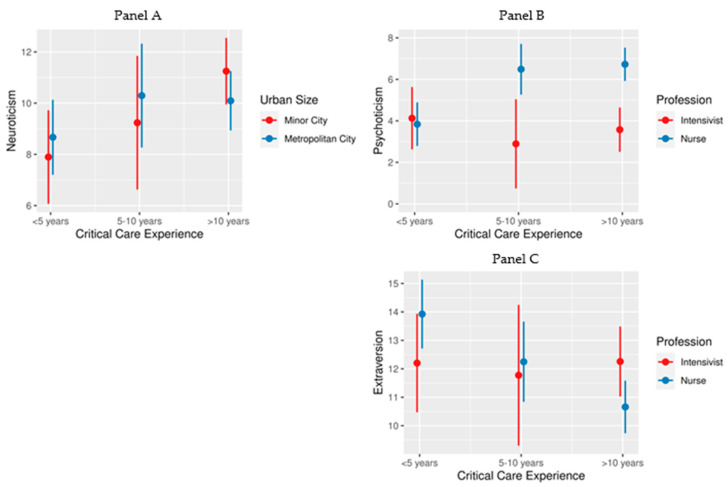
Interaction effects of critical care experience, urban size of ICU location, and profession on the personality traits in the context of the examined model. Notes: The length of critical care experience was categorized into three groups: less than 5 years, 5–10 years, and more than 10 years. (**Panel A**) Differences in neuroticism were assessed across these groups by the urban size of ICU location categorized as ‘minor city’ and ‘metropolitan city’. Differences in psychoticism (**Panel B**) and extraversion (**Panel C**) were assessed across these groups by the profession categorized as ‘intensivist’ and ‘nurse’. Each dot represents the mean score for the respective group, while the error bars signify 95% confidence intervals.

**Figure 3 healthcare-12-00587-f003:**
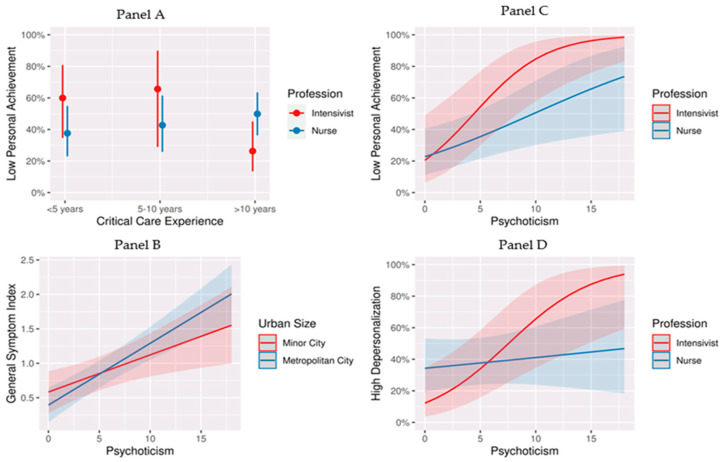
Interaction effects of critical care experience, profession, and urban size of ICU location on the probability of burnout symptoms and psychopathology in the context of the examined model. Notes: (**Panel A**) The probability of low personal achievement across different levels of critical care experience for the two professions. Each dot represents the mean score for the respective group, while the error bars signify 95% confidence intervals. (**Panel B**) The psychopathology scores against levels of psychoticism for individuals in metropolitan versus minor cities. The shaded areas around the lines represent 95% confidence intervals, while the lines show that as psychoticism increased, so did the psychopathology score, with a stronger effect seen in metropolitan areas. (**Panel C**) The relationship between the personality trait of psychoticism and the probability of low personal achievement for intensivists and nurses. The shaded areas around the lines represent 95% confidence intervals. The upward trend for both lines suggests that as psychoticism increased, the likelihood of low personal achievement also increased, with a steeper slope for intensivists than for nurses. (**Panel D**) The probability of experiencing high depersonalization against the level of psychoticism for intensivists and nurses. The shaded areas around the lines represent 95% confidence intervals. The trend indicates that higher levels of psychoticism were associated with a higher probability of experiencing high depersonalization. The difference in slopes between the two professions suggests that the effect of psychoticism on depersonalization was more pronounced in intensivists.

**Table 1 healthcare-12-00587-t001:** Correlation between personality traits and burnout facets of the examined sample.

			Personality Traits			Burnout Facets		
		a	Extraversion	Neuroticism	Psychoticism	Emotional Exhaustion	Depersonalization	Personal Achievement
Personality Traits	Extraversion	0.871						
	Neuroticism	0.951	−0.127 **					
	Psychoticism	0.907	−0.180 **	0.225 **				
Burnout facets	Emotional exhaustion	0.839	−0.131 **	0.370 **	0.133 **			
	Depersonalization	0.767	−0.078	0.297 **	0.277 **	0.624 **		
	Personal achievement	0.828	0.345 **	−0.167 **	−0.349 **	−0.045	−0.128 **	
Global severity index (GSI)		0.99	−0.168 **	0.476 **	0.474 **	0.512 **	0.431 **	−0.197 **

Notes: a = Cronbach’s alpha, ** correlation significant at 0.01 level.

**Table 2 healthcare-12-00587-t002:** Personality, burnout, and psychopathology of the examined sample.

Scale	Total (N = 503)	Profession		Sex		Critical Care Experience			Urban Size of ICU Location	
		Intensivist (n = 155)	Nurse (n = 348)	Male (n = 164)	Female (n = 339)	<5 Years (n = 166)	5–10 Years (n = 105)	10–15 Years (n = 232)	Metropolitan (n = 309)	Other (n = 194)
Age	41.5 (8.4)	45.7 (8.1)	39.6 (7.8)	43.0 (8.8)	41.0 (8.1)	35 (6.4)	42 (7.0)	46 (7.1)	41.6 (8.5)	41.4 (8.2)
Personality traits										
Psychoticism	5.8 (3.9)	4.5 (3.2)	6.3 (4.0)	6.3 (4.3)	5.5 (3.7)	4.9 (3.1)	6.6 (4.5)	5.9 (4.0)	5.8 (4)	5.7 (3.8)
Extraversion	12.2 (4.3)	12.1 (4.26)	12.2 (4.3)	12.1 (4.2)	12.2 (4.3)	13.2 (4.2)	12.0 (4.2)	11.0 (4.2)	12.2 (4.2)	12.1 (4.4)
Neuroticism	10.7 (4.5)	9.85 (4.62)	11.1 (4.4)	10.2 (4.8)	10.9 (4.4)	9.6 (4.3)	11.0 (4.7)	11.0 (4.4)	10.5 (4.3)	10.9 (4.9)
Burnout facets										
Emotional exhaustion	25.6 (11.1)	24.4 (10.8)	26.2 (11.2)	24.8 (11.6)	26 (10.8)	23.7 (10.3)	25.0 (11.0)	27.0 (11.0)	26 (11.2)	25 (10.9)
Depersonalization	10.4 (6.6)	10.1 (7.0)	10.6 (6.4)	10.8 (7.0)	10.3 (6.4)	9.5 (6.3)	10.0 (6.5)	11.0 (6.7)	10.8 (6.6)	9.9 (6.6)
Personal achievement	31.2 (8.7)	33.7 (7.1)	30.1 (9.1)	32.2 (8.4)	30.7 (8.8)	32.5 (7.7)	30.0 (10.0)	31.0 (8.7)	31.7 (8.4)	30.5 (9.1)
Global severity index (GSI)	0.89 (0.72)	0.72 (0.57)	0.96 (0.77)	0.84 (0.81)	0.91 (0.67)	0.71 (0.60)	1.1 (0.87)	0.93 (0.70)	0.92 (0.73)	0.85 (0.70)

Notes: N = total sample size (intensivists and nurses), n = sample size per group, ICU = intensive care unit. Means and standard deviations are presented.

**Table 3 healthcare-12-00587-t003:** Concurrence among burnout facets of the examined sample.

Emotional Exhaustion	Depersonalization	Personal Achievement		
		Low (≤29)	Moderate (30–38)	High (≥39)
Low (≤16)	Low (≤4)	20 (12/8) ^(1)^	16 (5/11)	9 (4/5)
	Moderate (5–12)	13 (7/6)	14 (7/7)	27 (2/25)
	High (≥13)	5 (2/3)	1 (0/1)	0 (0/0)
Moderate (17–33)	Low (≤4)	21 (8/13)	25 (11/14)	10 (2/8)
	Moderate (5–12)	20 (9/11)	52 (11/41)	48 (8/40)
	High (≥13)	13 (2/11)	34 (13/21)	46 (14/32)
High (≥34)	Low (≤4)	2 (1/1)	2 (1/1)	0 (0/0)
	Moderate (5–12)	2 (1/1)	11 (4/7)	23 (3/20)
	High (≥13)	19 (5/14)	38 (13/25)	32 (10/22)

Notes: ^(1)^ frequencies of total sample (frequencies of intensivists/frequencies of nurses).

**Table 4 healthcare-12-00587-t004:** Moderation analysis of personality dimensions of the examined sample.

	Neuroticism ^(1)^				Psychoticism ^(2)^				Extraversion ^(3)^			
	B	*p*	95% C.I.		B	*p*	95% C.I.		B	*p*	95% C.I.	
			Lower	Upper			Lower	Upper			Lower	Upper
Constant	5.283	0.028	0.562	10.00	5.157	0.009	1.275	9.039	11.07	0.000	6.600	15.55
Critical Care Experience												
D_1_: 5–10 vs. <5	1.364	0.700	−5.598	8.326	−5.543	0.058	−11.27	0.180	0.522	0.877	−6.073	7.116
D_2_: 10–15 vs. 5–10	4.231	0.194	−2.155	10.62	1.642	0.539	−3.608	6.892	2.152	0.485	−3.896	8.201
Profession	**1.693**	**0.038**	**0.091**	**3.294**	−0.289	0.666	−1.606	1.027	**1.721**	**0.026**	**0.204**	**3.238**
Critical Care Experience × Profession												
D_1_ × Profess.	−0.311	0.824	−3.065	2.442	**3.883**	**0.001**	**1.620**	**6.147**	−1.246	0.348	−3.855	1.362
D_2_ × Profess.	−0.001	0.999	−2.556	2.554	−0.444	0.678	−2.545	1.656	**−2.074**	**0.093**	**−4.494**	**0.346**
Urban size of ICU Location	0.771	0.287	−0.651	2.194	0.370	0.534	−0.799	1.540	−0.325	0.636	−1.672	1.022
Critical Care Experience × Uban Size of ICU Location												
D_1_ × Urban Size of ICU Location	0.288	0.808	−2.040	2.615	0.424	0.663	−1.489	2.337	0.295	0.793	−1.909	2.499
D_2_ × Urban Size of ICU Location	**−2.216**	**0.046**	**−4.392**	**−0.040**	−0.515	0.572	−2.304	1.273	0.406	0.699	−1.655	2.467
Age	0.004	0.907	−0.057	0.064	−0.027	0.289	−0.076	0.023	−0.006	0.824	−0.064	0.051
Sex	−0.199	0.647	−1.054	0.655	**1.421**	**0.000**	**0.719**	**2.123**	−0.246	0.550	−1.055	0.563

Notes: D = duration of critical care experience based on years of experience, B = unstandardized estimates, *p* = *p*-value, C.I. = confidence interval, ICU = intensive care unit, R^2^ = R-squared measure that represents the proportion of the variance in the dependent variable, significant differences are marked with bold, ^(1)^ R^2^ = 0.067, F (10, 492) = 3.531, *p* < 0.001. ^(2)^ R^2^ = 0.147, F (10, 492) = 8.491, *p* < 0.001. ^(3)^ R^2^ = 0.062, F (10,492) = 3.342, *p* < 0.001.

**Table 5 healthcare-12-00587-t005:** Moderation analysis of burnout and psychopathology of the examined sample.

	High Emotional Exhaustion ^(1)^				High Depersonalization ^(2)^				Low Personal Achievement ^(3)^				Global Severity Index (GSI) ^(4)^			
	B	*p*	95% C.I.		B	*p*	95% C.I.		B	*p*	95% C.I.		B	*p*	95% C.I.	
			Lower	Upper			Lower	Upper			Lower	Upper			Lower	Upper
Constant	−0.571	0.840	−6.126	4.984	−2.857	0.277	−8.006	2.292	0.130	0.962	−5.190	5.451	−0.367	0.578	−1.662	0.929
Socio-demographics																
Age	**0.037**	**0.027**	**0.004**	**0.071**	0.017	0.265	−0.013	0.047	**−0.050**	**0.004**	**−0.084**	**−0.015**	**0.009**	**0.020**	**0.001**	**0.017**
Sex	0.135	0.588	−0.353	0.623	0.018	0.937	−0.422	0.458	−0.449	0.070	−0.934	0.037	−0.070	0.230	−0.183	0.044
Critical Care Experience																
D_1_: 5–10 vs. <5	−1.365	0.532	−5.644	2.914	−2.395	0.249	−6.470	1.681	0.434	0.832	−3.565	4.432	−0.174	0.712	−1.102	0.753
D_2_: 10–15 vs. 5–10	0.938	0.626	−2.835	4.712	2.438	0.194	−1.244	6.119	−4.318	0.025	−8.092	−0.545	−0.113	0.790	−0.950	0.723
Profession	−0.904	0.431	−3.153	1.345	0.261	0.804	−1.800	2.322	0.944	0.387	−1.196	3.085	0.132	0.614	−0.381	0.645
D_1_ × Profession	−0.138	0.869	−1.772	1.496	0.736	0.399	−0.975	2.448	−0.031	0.970	−1.619	1.558	−0.005	0.976	−0.371	0.360
D_2_ × Profession	0.485	0.514	−0.972	1.942	−1.356	0.092	−2.934	0.222	**1.967**	**0.012**	**0.433**	**3.501**	0.062	0.714	−0.272	0.397
Urban Size of ICU Location	−1.472	0.172	−3.585	0.642	−0.225	0.813	−2.091	1.641	0.060	0.950	−1.842	1.963	−0.178	0.468	−0.658	0.303
D_1_ × Urban Size of ICU Location	0.708	0.355	−0.791	2.208	0.554	0.368	−0.652	1.760	−0.160	0.803	−1.418	1.098	0.151	0.338	−0.159	0.462
D_2_ × Urban Size of ICU Location	−0.664	0.329	−1.998	0.669	0.238	0.672	−0.863	1.339	0.674	0.260	−0.499	1.846	−0.068	0.639	−0.355	0.218
Personality Traits																
P	0.229	0.189	−0.113	0.571	**0.455**	**0.014**	**0.092**	**0.818**	**0.488**	**0.014**	**0.098**	**0.877**	0.029	0.502	−0.056	0.114
E	−0.234	0.105	−0.518	0.049	−0.207	0.121	−0.470	0.055	−0.224	0.148	−0.528	0.080	−0.019	0.578	−0.085	0.048
N	−0.039	0.782	−0.315	0.237	0.168	0.191	−0.084	0.421	**0.327**	**0.033**	**0.026**	**0.627**	**0.082**	**0.011**	**0.019**	**0.145**
P × Profession	−0.099	0.147	−0.233	0.035	**−0.233**	**0.003**	**−0.388**	**−0.077**	**−0.182**	**0.029**	**−0.346**	**−0.019**	−0.011	0.529	−0.045	0.023
E × Profession	0.060	0.311	−0.056	0.175	0.098	0.072	−0.009	0.205	0.009	0.885	−0.113	0.131	−0.001	0.928	−0.028	0.026
N × Profession	0.066	0.237	−0.044	0.176	−0.012	0.814	−0.113	0.088	−0.086	0.162	−0.206	0.034	−0.001	0.911	−0.027	0.024
P × Urban Size of ICU Location	−0.019	0.759	−0.141	0.103	0.039	0.487	−0.071	0.150	0.001	0.982	−0.120	0.123	**0.036**	**0.017**	**0.006**	**0.065**
E × Urban Size of ICU Location	0.087	0.112	−0.021	0.195	0.022	0.658	−0.075	0.119	0.045	0.422	−0.064	0.154	0.009	0.476	−0.016	0.034
N × Urban Size of ICU Location	0.037	0.504	−0.071	0.145	−0.044	0.368	−0.139	0.051	−0.088	0.098	−0.193	0.016	−0.012	0.354	−0.036	0.013

Notes: D = duration of critical care experience based on years of experience, D_1_: 5–10 vs. <5, D_2_: 10–15 vs. 5–10, B = unstandardized estimates, *p* = *p*-value, C.I. = confidence interval, ICU = intensive care unit, R^2^ = R-squared measure that represents the proportion of the variance in the dependent variable, P = psychoticism, E = extraversion, N = neuroticism, significant differences are marked with bold, ^(1)^ R^2^ = 0.156, c2(19) = 56.298, *p* < 0.001. ^(2)^ R^2^ = 0.157, c2(19) = 61.324, *p* < 0.001. ^(3)^ R^2^ = 0.321, c2(19) = 135.995, *p* < 0.001. ^(4)^ R^2^ = 0.398, F (19, 483) = 16.837, *p* < 0.001.

## Data Availability

The data presented in this study are available upon request from the corresponding author.

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
