# Peer review of "Personality Traits, Burnout, and Psychopathology in Healthcare Professionals in Intensive Care Units—A Moderated Analysis"

_healthcare, 2024, doi:10.3390/healthcare12050587_

Round 1
Reviewer 1 Report
Comments and Suggestions for Authors
Dear authors,
Congratulations on the article developed.
The study is pertinent and relevant, having high potential to contribute to personalized support systems for intensivists and nurses in the ICU context.
The article is well written overall, but needs slight reformulations. I leave some suggestions for improvement for further analysis.
Title:
Very long. It must be up to 16 words. I suggest removing “Exploring the Nexus:” and aligning it with the keywords, including the words health professionals and Intensive Care Unit.
Suggestion only (it can be different, or with another syntax, as long as it includes all relevant terms):
Personality Traits, Burnout and Psychopathology in healthcare professionals in Intensive Care Units – A Moderated Analysis
Summary:
Put ICU in full, followed by the abbreviation, the first time you mention the term. Put EPQ in full.
Keywords: Put ICU in full.
Introduction:
Line 51 - Put ICU in full, followed by the abbreviation, the first time the term is mentioned in the main text (and not on line 101).
Method: Overall, well exposed and reproducible.
Results:
Place at the end of all tables the caption of all acronyms and abbreviations used in the tables.
Line 184: standardize “of the examined sample” in all tables (include or exclude in all)
Line 186: (1) should be raised and not as a normal number.
Line 187: put GSI in parentheses, after the first time the term “General Symptom Index” appears
Lines 201 and 202: the data reported here are not represented in figure 2.
Line 202: full stop missing.
Lines 204 and 205: (1) (2) and (3) should be raised and not as a normal number.
A number (4) appears in table 4 that has no correspondence.
Line 207: name from figure 2 – standardize name used Intensive Care experience or Critical care experience.
Discussion: Overall, very good.
Conclusions: Overall, good.
References:
Line 489: This reference is not cited and is underlined.
Confirm that all references in the list are cited.
Check, throughout the text, whether the citations correspond correctly to the references.
Best wishes for the future!
Author Response
Dear authors,
Congratulations on the article developed.
The study is pertinent and relevant, having high potential to contribute to personalized support systems for intensivists and nurses in the ICU context.
The article is well written overall, but needs slight reformulations. I leave some suggestions for improvement for further analysis.
- We thank the reviewer.
Title: Very long. I suggest removing “Exploring the Nexus:” and aligning it with the keywords, including the words health professionals and Intensive Care Unit. Suggestion only (it can be different, or with another syntax, as long as it includes all relevant terms): Personality Traits, Burnout and Psychopathology in healthcare professionals in Intensive Care Units – A Moderated Analysis
- We agree with the reviewer, and we have now amended the manuscript's title to "Personality Traits, Burnout, and Psychopathology in Healthcare Professionals in Intensive Care Units – A Moderated Analysis” as per requested (Page 1, now lines 4-6).
Summary: Put ICU in full, followed by the abbreviation, the first time you mention the term.
- Done (Page 1, now line 15).
Put EPQ in full.
- Done (Page 1, now lines 21-22).
Keywords: Put ICU in full.
- Done (Page 1, now line 35).
Introduction: Line 51 - Put ICU in full, followed by the abbreviation, the first time the term is mentioned in the main text (and not on line 101).
- Done (Page 2, now line 54).
Method: Overall, well exposed and reproducible.
- We thank the reviewer.
Results:
Place at the end of all tables the caption of all acronyms and abbreviations used in the tables.
- Done, we have now placed all acronyms and abbreviations used in the tables (Pages 6, 7, 8, and 10).
Line 184: standardize “of the examined sample” in all tables (include or exclude in all)
- We have now added this statement in all tables. Pages 6, 7, and 8.
Line 186: (1) should be raised and not as a normal number.
- Done, number (1) is now has been raised. Page 7, now line 261.
Line 187: put GSI in parentheses, after the first time the term “General Symptom Index” appears.
- Done (Page 5, now line192, which is the first time use the term). We have also changed the term as Global Severity Index, as originally reported by the developer of the scale (Derogatis, L.R. (1997). SCL-90R: Administration, scoring and procedures manual for the revised version. John Hopkins University, Baltimore).
Lines 201 and 202: the data reported here are not represented in figure 2.
- We agree with the reviewer, and we have now added that ‘A noteworthy main effect of profession on neuroticism emerged, indicating that nurses exhibited higher levels of neuroticism in comparison to intensivists (M = 11.1 vs M = 9.9, p = 0.038’, indicating the major results of main effects analysis. Page 7, now lines 278-280.
- We also added that “Furthermore, our analysis showed some significant interaction effects (Table 2), which are further illustrated in Figure 2, Panels A, B, and C. A noteworthy interaction be-tween critical cere experience and urban size of ICU location concerning neuroticism was observed, suggesting that ICU personnel in minor cities tend to report higher levels of neuroticism across all experience levels than in metropolitan cities, with the greatest difference seen in the group with more than 10 years of experience (Figure 2, Panel A). Additionally, the analysis revealed a significant interaction between critical care experience and profession regarding psychoticism and extraversion. Specifically, as experience in the ICU increases, nurses exhibited higher levels of psychoticism and lower levels of extraversion (Figure 2, Panels B and C).” Page 8, now lines 286-295).
Line 202: full stop missing.
- Done (Page 8, now line 295).
Lines 204 and 205: (1) (2) and (3) should be raised and not as a normal number.
- Done (Page 8, now lines 283, 284).
A number (4) appears in table 4 that has no correspondence.
- Number (4) was written in error and now has been removed. Page 8, Table 4.
Line 207: name from figure 2 – standardize name used Intensive Care experience or Critical care experience.
- Thank you, we have now utilized critical care experience throughout the text, tables, and figures. The changes are marked Word's track changes feature.
Discussion: Overall, very good.
Conclusions: Overall, good.
- We thank again the reviewer.
References:
Line 489: This reference is not cited and is underlined.
- Done, the reference is now cited in the text. Page 3, now line 120.
Confirm that all references in the list are cited. Check, throughout the text, whether the citations correspond correctly to the references.
- Thank you, we have reviewed all the references in the text and the list, and we have made the necessary modifications accordingly. The new references are colored with yellow.
Best wishes for the future!
- We thank again the reviewer.
# Please see attached file for cover letter with point to point response to all reviewers comments

Reviewer 2 Report
Comments and Suggestions for Authors
This study investigates the connections between personality traits, burn-out, and psychopathology among Intensive Care Unit (ICU) healthcare professionals with a focus on professional roles (nurses vs. intensivists) and hospital locations (metropolitan vs. smaller cities). I recommend several points that need to be considered in your article. I hope it might help you with the improvement.
The title is hard to follow and needs to be shorter and more precise.
Please rephrase lines 12-14
The authors should consider other wards for burn-out issues, e.g., psychiatry (line 50).
The authors should consider the issue of healthcare workers' aggression, which can lead to burn-out. Nurses and intensivists, as well as psychiatrists, are very often involved (line 60; please see: DOI: 10.1097/NMD.0000000000001504)
Please better explain the PEN model in line 72.
Please enrich the introduction with worldwide information regarding the number of burn-out issues among healthcare professionals.
Lines 140-145: please specify if all the tools were validated in Greek.
Please describe the statistical analysis better and if a statistician evaluated the test.
Please integrate Figure 2 with all the other analyses to clarify the results more; moreover, explain better the statistical analysis used in the note.
Please rephrase lines 227-236
Figure 3: Please provide a more exhaustive explanation of the figure.
Lines 242-246: this consideration should be replaced in the discussion section.
Lines: the phrase in 292-293 contrasts with that in 60-62.
Please provide to insert references in line 304.
Please explain better the phrase in lines 313-319.
Please argue the phrase in lines 330-337 and provide the references.
Moreover, the manuscript lacks any mention or reference to the consequences of this issue on the care provided to the patients. Furthermore, the study's limitations section briefly touches upon prospects without offering concrete proposals. As such, revisiting and expanding upon these areas may be beneficial to provide a more comprehensive and informative analysis. Please provide.
Author Response
REVIEWER 2
This study investigates the connections between personality traits, burn-out, and psychopathology among Intensive Care Unit (ICU) healthcare professionals with a focus on professional roles (nurses vs. intensivists) and hospital locations (metropolitan vs. smaller cities). I recommend several points that need to be considered in your article. I hope it might help you with the improvement.
- We thank the reviewer.
The title is hard to follow and needs to be shorter and more precise.
- As per answer to Reviewer 1, we have now amended the manuscript's title to "Personality Traits, Burnout, and Psychopathology in Healthcare Professionals in Intensive Care Units – A Moderated Analysis” as per requested (Page 1, now lines 4-6).
Please rephrase lines 12-14
- We have now modified these lines as follows: “This study explored the associations between personality dimensions, burnout, and psychopathology in healthcare professionals in Intensive Care Units (ICUs). The study further aimed to discern the differences in these relationships when considering the variables of critical care experience (less than 5 years, 5-10 years, and more than 10 years), profession (nurses versus intensivists), and the urban size of the city where the ICU is located (metropolitan cities versus smaller urban cities)”. Page 1, now lines 14-19.
- We have also made some corrections in the whole abstract for consistency of the terminology used and clarity.
The authors should consider other wards for burn-out issues, e.g., psychiatry (line 50).
- We have now added that “Mental health professionals are also susceptible to high levels of burnout syndrome (O'Connor et al., 2018) due to the significant emotional and cognitive stress associated with their profession (Jørgensen, et al., 2021).’’ Page 2, now lines 55-57. Additional references are added to the list.
The authors should consider the issue of healthcare workers' aggression, which can lead to burn-out. Nurses and intensivists, as well as psychiatrists, are very often involved (line 60; please see: DOI: 10.1097/NMD.0000000000001504)
- We thank the reviewer we have now added that “A common factor across all professions may be the problem of aggression faced by healthcare workers. This aggression, whether it comes from patients or workplace interactions, is considered a precursor to burnout (Flannery, et al., 2011; Mento, et al., 2020; Mele et al., 2022)”. Page 2, now lines 68-71. Additional references are added to the list.
Please better explain the PEN model in line 72.
- We agree with the reviewer. We have now added that “Utilizing the dimensions of psychoticism (P), extraversion (E), and neuroticism (N), also known as Eysenck’s PEN model (Eysenck & Eysenck, 1975), this study defines personality through these three dimensions. The PEN model posits that these dimensions represent fundamental aspects of an individual's personality, influencing their behavior, cognition, and emotional responses (Eysenck & Eysenck, 1971; Maher, & Maher, 1994). Specifically, psychoticism is associated with traits such as disagreeableness, non-conscientiousness, and a propensity for risk-taking. Extraversion encompasses characteristics like sociability, assertiveness, positive emotions, and impulsivity, while neuroticism involves traits such as anxiety, depression, and self-doubt. Furthermore, these personality dimensions—P, E, and N—are considered orthogonal (Eysenck & Eysenck, 1971, indicating no correlation between them, thus contributing to a comprehensive understanding of various facets of an individual's personality (Eysenck & Eysenck, 1985).” Pages 2-3, now lines 90-101. Additional references are added to the list.
Please enrich the introduction with worldwide information regarding the number of burn-out issues among healthcare professionals.
- We have now added that “Burnout has emerged as a pervasive issue among healthcare professionals (Maslach & Leiter, 2016). For example, Woo et al. (2020) found that 11.23% of nurses globally have experienced symptoms of burnout. In the field of emergency medicine, healthcare workers exhibit a burnout prevalence rate of 20-60% (Kimo Takayesu et al., 2014). Similarly, 40% of mental health professionals have encountered burnout (O'Connor et al., 2018), and Rotenstein et al. (2018) report that 67% of physicians have demonstrated overall burnout prevalence estimates” Page 2, now lines 72-78. Additional references are added to the list.
Lines 140-145: please specify if all the tools were validated in Greek.
- As previously mentioned on Page 4, lines140-145, participants were required to complete the Greek-adapted versions of the Maslach Burnout Inventory (MBI) (Anagnostopoulos & Papadatou, 1992), the Eysenck Personality Questionnaire (EPQ) (Dimitriou, 1986), and the Symptom Checklist-90 (SCL-90) (Donias et al., 1991).
- However, as per answer to Reviewer 3, we have now provided brief description of the instruments used. Specifically, we have added that: The MBI developed by Maslach and Jackson (1981), assesses burnout through 22 items grouped into three subscales: emotional exhaustion, depersonalization, and per-sonal achievement. Respondents rate each item on a 7-point Likert scale ranging from ‘never’ (0) to ‘daily’ (6). The scale has been translated into Greek and has shown satisfactory validity and reliability (Anagnostopoulos & Papadatou, 1992). In this study, we used the established thresholds on the tree subscales, based on the adapted Greek version by Zis et al. (2014). Specifically, burnout was identified for scores ≥34 in emotional exhaustion, ≥13 in depersonalization, and ≤29 in personal accomplishment. The EPQ developed by Eysenck and Eysenck (1975), assesses personality traits using a 90-item questionnaire. It measures three primary dimensions: extraversion, neuroticism, and psychoticism. Respondents rate each item on a binary scale (yes/no response). The scale has been translated into Greek and has shown satisfactory validity and reliability (Dimitriou, 1986). The SCL-90, developed by Derogatis (1977), assesses a wide range of psychological symptoms. It comprises 90 questions across 9 subscales, namely somatization, obsessive-compulsive, interpersonal sensitivity, depression, anxiety, hostility, phobic anxiety, paranoid ideation, and psychoticism. Participants rate items on a 5-point Likert scale ranging from 0 (Not at all) to 4 (Extremely), based on distress level. Additionally, three global indices (Global Severity Index [GSI], Positive Symptom Distress Index [PSDI], and Positive Symptom Total [PST]) summarize psychological distress. The scale has been translated into Greek and validated for criterion and convergent validity (Donias et al., 1991). This study reports data only for the GSI index, a measure that reflects the overall psychopathology. A higher GSI score indicates higher levels of psychological distress. Page 5, now lines 174-197.
Please describe the statistical analysis better and if a statistician evaluated the test.
- We thank the reviewer. We have now modified the analysis as follows: ‘Initial data analysis involved summarizing measures of central tendency and dispersion, with means and standard deviations reported for continuous variables, and pro-portions for categorical ones. To compare groups, we employed independent samples t-tests, chi-square tests of independence, and one-way ANOVA. To evaluate the internal consistency and reliability of the utilized scales - namely, the MBI, the EPQ, and the SCL-90 - a Cronbach's alpha analysis was conducted (Cronbach & Meehl, 1995). Additionally, a Pearson correlation analysis was executed to examine the correlation matrix, encompassing age, personality traits, burnout, and psychopathology. To evaluate the theoretical model, a moderated mediation approach was employed, wherein the relationship between critical care experience and burnout was mediated by three personality dimensions, while profession moderated this relationship (Figure 1). Model 76 of the PROCESS function for R, developed by Hayes (2018), was utilized four times: once for each facet of burnout (i.e.., emotional exhaustion, depersonalization, and personal achievement) and once for the psychopathology, as indicated by the GSI. This model was chosen for its robust capabilities in handling moderated mediation models. Furthermore, we categorized critical care experience into three groups: less than 5 years, 5-10 years, and more than 10 years. Each group was compared to the preceding group using a sequential coding scheme. This approach facilitated the comparison of successive levels of experience and enabled a nuanced exploration of the progressive nature of phenomena within the ICU setting. In both the moderation analysis of personality dimensions and the moderation analysis of burnout and psychopathology, main effects and interaction effects were examined. Bootstrap analysis was conducted to evaluate the indirect effects of the theoretical model. Unstandardized coefficients (B) and 95% confidence intervals were reported. Additionally, we reported the R² for each model to assess the proportion of variance explained. Significant interaction effects were graphically illustrated. The statistical analysis was conducted under the guidance of a statistician to ensure accuracy and rigor in the interpretation of results. A two-sided alpha level of 0.05 was set for determining statistical significance. All data were analyzed using R statistical language (R Core Team, 2021) equipped with PROCESS function (Hayes, 2018). Pages 5-6, now lines 203-233.
Please integrate Figure 2 with all the other analyses to clarify the results more; moreover, explain better the statistical analysis used in the note.
- We appreciate the reviewer's input and the chance to elucidate our findings further. The analyses depicted in Figure 2 are a visual extension of the results presented in Table 2, focusing particularly on the interaction effects identified. Figure 2 serves to graphically illustrate these interactions for a clearer understanding. We have now added that “Furthermore, our analysis showed some significant interaction effects (Table 2), which are further illustrated in Figure 2, Panels A, B, and C. A noteworthy interaction be-tween critical cere experience and urban size of ICU location concerning neuroticism was observed, suggesting that ICU personnel in minor cities tend to report higher levels of neuroticism across all experience levels than in metropolitan cities, with the greatest difference seen in the group with more than 10 years of experience (Figure 2, Panel A). Additionally, the analysis revealed a significant interaction between critical care experience and profession regarding psychoticism and extraversion. Specifically, as experience in the ICU increases, nurses exhibited higher levels of psychoticism and lower levels of extraversion (Figure 2, Panels B and C)’. Page 8, now lines 286-295.
- In addition, we added the following notes under Figure 2: ‘The length of critical care experience is categorized into three groups: less than 5 years, 5-10 years, and more than 10 years. Differences in neuroticism, were assessed across these groups by the urban size of ICU location categorized as with 'Minor City' and 'Metropolitan City'. Differences in psychoticism and extraversion were assessed across these groups by the profession categorized as 'Intensivist' and 'Nurse'. Each dot represents the mean score for the respective group, while the error bars signify 95% confidence intervals’. Page 9, now lines 299-304.
Please rephrase lines 227-236
- We thank the reviewer. We have now rephased the above lines as follows: ‘) Additionally, the analysis revealed significant interaction effects (Table 5) between psychoticism, critical care experience, profession, and the urban size of the ICU location, which are further illustrated in Figure 3, Panels A, B, C, and D. Notably, with increasing duration of critical care experience, nurses tend to report lower personal achievement scores, whereas intensivists show an opposite trend (Figure 3, Panel A). Moreover, the relationship between psychoticism and psychopathology, as measured by the GSI, varied in strength depending on the urban size of the ICU location. Specifically, ICU personnel in larger metropolitan areas exhibited a more robust relationship between psychoticism and GSI scores than those in smaller urban settings (Figure 3, Panel B). Lastly, there was a significant association between psychoticism and both low personal achievement and high depersonalization scores. This relationship was particularly strong among intensivists when compared to nurses (Figure 3, Panels C and D)’ Page 10, now lines 324-335.
Figure 3: Please provide a more exhaustive explanation of the figure.
- We have now added the following notes for Figure 3: ‘Panel A illustrates the probability of low personal achievement across different levels of critical care experience for the two professions. Each dot represents the mean score for the respective group, while the error bars signify 95% confidence intervals. Panel B illustrates the psychopathology scores against levels of psychoticism for individuals in metropolitan versus minor cities. The shaded areas around the lines represent 95% confidence intervals, while the lines show that as psychoticism increases, so does the psychopathology score, with a stronger effect seen in metropolitan areas. Panel C illustrates the relationship between the personality trait of psychoticism and the probability of low personal achievement for intensivists and nurses. The shaded areas around the lines represent 95% confidence intervals. The upward trend for both lines suggests that as psychoticism increases, the likelihood of low personal achievement also increases, with a steeper slope for intensivists than for nurses. Panel D illustrates the probability of experiencing high de-personalization against the level of psychoticism for intensivists and nurses. The shaded areas around the lines represent 95% confidence intervals. The trend indicates that higher levels of psychoticism are associated with a higher probability of experiencing high depersonalization. The difference in slopes between the two professions suggests that the effect of psychoticism on depersonalization is more pronounced in intensivists.’ Page 11, now lines 340-335.
Lines 242-246: this consideration should be replaced in the discussion section.
- We agree with the reviewer. We have now modified the statement as follows: ‘In the context of the examined model, bootstrap analysis showed that personality dimensions do not have a significant indirect effect on the relationship between critical care experience and both psychopathology and burnout’ Page 11, now lines 355-357.This decision was made to present the results of the bootstrap analysis in a straightforward manner, with the explanations provided in the discussion section.
- We added this consideration as per requested in the discussion section as follows: ‘Finally, this study found that personality traits do not significantly influence the relationship between critical care experience, psychopathology, and burnout. Essentially, the direct effects of critical care experience on both psychopathology and burnout are substantial enough to explain their relationships without the need to consider personality traits as an explanatory factor. This suggests that the impact of critical care experience on psychopathology and burnout occurs independently of an individual's personality traits.’ Page 14, now lines 485-490.
Lines: the phrase in 292-293 contrasts with that in 60-62.
- We agree with the reviewer, however as we have stated in the introduction the prevalence of burnout in the ICU setting also shows significant variability (Chuang et al., 2016; Van Mol et al., 2015). We have now added the following statement to highlight this dispensary. ‘However, this stands in contrast to other studies (Dulko & Zangaro, 2022; Ntantana et al., 2017). This discrepancy in findings might be attributable to the specific samples and settings of the various studies.’ Page 12, now lines 405-407.
Please provide to insert references in line 304.
- Despite the absence of directly relevant references to our knowledge and the speculative nature of the statements in those lines, we have included the following reference: Leiter, M. P., & Spence Laschinger, H. K. (2006). Relationships of work and practice environment to professional burnout: testing a causal model. Nursing research, 55(2), 137–146. https://doi.org/10.1097/00006199-200603000-00009. Page 12, now line 417.
Please explain better the phrase in lines 313-319.
- We have now modified the statement as follows: ‘Interestingly, a positive correlation was found between age and personal achievement among ICU personnel, with increasing age associated with a lower likelihood of re-porting low personal accomplishment. This trend suggests that with advancing age, there is a discernible decrease in reporting low levels of personal accomplishment, indicating that older ICU personnel may perceive their professional achievements more positively. This can largely be attributed to the cumulative effect of accrued experience over time in relation to job satisfaction (Carrillo-García, et al, 2013), which ostensibly contributes to a more favorable self-assessment of accomplishments in the challenging ICU environment. Therefore, the career trajectory of healthcare professionals in such settings might be characterized by a gradual and steady increase in personal accomplishment, underpinned by the development of adaptive skills and psychological resilience. In contrast, there was a significant negative correlation between age and emotional exhaustion and psychopathology, confirming a general trend towards an increased feeling of emotional exhaustion in older ICU personnel, as reported in various studies (Padilla Fortunatti & Palmeiro-Silva, 2017; Poncet et al., 2007; Kashtanov et al., 2022).’ Page 13, now lines 427-441.
Please argue the phrase in lines 330-337 and provide the references.
- We have now added that ‘This amplification could be particularly pronounced in high-stress healthcare settings, where the additional urban stressors can intensify the psychological impact of already demanding jobs. Studies have shown that environmental factors like urban living conditions can significantly affect mental health, potentially exacerbating underlying personality traits (Lederbogen et al., 2011; Peen et al., 2010). Additionally, the role of organizational culture in hospitals, especially in metropolitan areas, is critical. A less supportive work environment and limited mental health resources can leave ICU personnel, particularly those with higher psychoticism scores, more vulnerable to psychological distress and less equipped to cope effectively. The importance of organizational support in mitigating the impact of stress on mental health has been emphasized in various studies (Shanafelt et al., 2012; West et al., 2009)’. Page 13, now lines 455-465. Additional references are added to the list.
Moreover, the manuscript lacks any mention or reference to the consequences of this issue on the care provided to the patients. Furthermore, the study's limitations section briefly touches upon prospects without offering concrete proposals. As such, revisiting and expanding upon these areas may be beneficial to provide a more comprehensive and informative analysis. Please provide.
- We agree with the reviewer. We have now added that ‘It is important to note that the repercussions of high levels of burnout and psychopathology among ICU personnel are significant, impacting both the healthcare professionals themselves (Moss et al., 2016) and the quality of patient care (De Hert, 2020). Research indicates that healthcare personnel experiencing burnout are associated with adverse patient outcomes. This includes diminished patient safety (Garcia et al., 2019; Ryu & Shim, 2021), increased standardized mortality ratios, and prolonged hospitalization durations (Welp et al., 2015). Additionally, there is a correlation between healthcare worker burnout and reduced patient satisfaction (Haas et al., 2000), as well as an increased incidence of medical errors and malpractice (De Hert, 2020’ Page 14, now lines 505-514.
- Moreover, we added a statement regarding study’s implications. We have stated that ‘Our study, therefore, lays a foundation for future explorations into strategic interventions aimed at establishing specialized training programs and robust support frameworks within the ICU. These programs and support systems are envisioned to offer both emotional and professional assistance to ICU personnel, thereby enhancing their ability to effectively manage the intense stress characteristic of the ICU and urban environment. Such proactive measures are crucial for maintaining the mental well-being and professional efficacy of ICU personnel’. Page 14, now lines 515-521.
- We have now offered some concrete proposals based on our limitations, in the relevant section. Specifically, we have added that ‘To enhance generalizability, future studies should include ICU personnel from various regions and cultural backgrounds. Conducting longitudinal research, as opposed to the cross-sectional approach used in this study, could offer insights into how these relationships evolve over time, particularly how personality traits interact with burnout and psychopathology in the long run. Additionally, comparing ICU personnel in urban versus rural settings, or in different healthcare systems, might reveal how environmental and systemic factors influence these relationships.’ Pages 14-15, now lines 527-534.
- In addition, we have added that ‘Exploring factors such as workplace environment, support systems, and personal life stressors could provide a more comprehensive understanding of the dynamics at play’ Page 15, now lines 542-543.
- At the end of the limitations section, we have also added ‘Finally, qualitative studies, such as interviews or focus groups, could provide deeper in-sights into the personal experiences and perspectives of ICU personnel, which might be missed in quantitative models. Page 15, now lines 546-549.
#in the attached word-file there is cover letter and the complete point-to-point response to all reviewers`comments

Reviewer 3 Report
Comments and Suggestions for Authors
Dear authors,
The manuscript addresses a relevant issue, is clearly written, provides sufficient background and relevant references.
The research design is appropriate, and the methods adequately described. Nevertheless, there could be a brief explanation of the instruments used to collect the data. Also, ethical concerns could be made more explicit when collecting and analyzing data.
Results are clearly presented, however, it is advisable to review the following aspects:
(i) In table 1, to clarify the meaning of "a", and to explain in line 169 how "internal reliability" was analyzed;
(ii) to revise the information presented in lines 201/202 and correspondence with figure 2.
The conclusions are well supported by the data, and the authors discuss the limitations of the study.
Overall, the manuscript publication is recommended.
Author Response
REVIEWER 3
Dear authors,
The manuscript addresses a relevant issue, is clearly written, provides sufficient background and relevant references.
- We thank the reviewer.
The research design is appropriate, and the methods adequately described. Nevertheless, there could be a brief explanation of the instruments used to collect the data. Also, ethical concerns could be made more explicit when collecting and analyzing data.
- We agree with the reviewer we have now added a brief description of the instruments used. Specifically, we have stated that “The MBI developed by Maslach and Jackson (1981), assesses burnout through 22 items grouped into three subscales: emotional exhaustion, depersonalization, and per-sonal achievement. Respondents rate each item on a 7-point Likert scale ranging from ‘never’ (0) to ‘daily’ (6). The scale has been translated into Greek and has shown satisfactory validity and reliability (Anagnostopoulos & Papadatou, 1992). In this study, we used the established thresholds on the tree subscales, based on the adapted Greek version by Zis et al. (2014). Specifically, burnout was identified for scores ≥34 in emotional exhaustion, ≥13 in depersonalization, and ≤29 in personal accomplishment. The EPQ developed by Eysenck and Eysenck (1975), assesses personality traits using a 90-item questionnaire. It measures three primary dimensions: extraversion, neuroticism, and psychoticism. Respondents rate each item on a binary scale (yes/no response). The scale has been translated into Greek and has shown satisfactory validity and reliability (Dimitriou, 1986). The SCL-90, developed by Derogatis (1977), assesses a wide range of psychological symptoms. It comprises 90 questions across 9 subscales, namely somatization, obsessive-compulsive, interpersonal sensitivity, depression, anxiety, hostility, phobic anxiety, paranoid ideation, and psychoticism. Participants rate items on a 5-point Likert scale ranging from 0 (Not at all) to 4 (Extremely), based on distress level. Additionally, three global indices (Global Severity Index [GSI], Positive Symptom Distress Index [PSDI], and Positive Symptom Total [PST]) summarize psychological distress. The scale has been translated into Greek and validated for criterion and convergent validity (Donias et al., 1991). This study reports data only for the GSI index, a measure that reflects the overall psychopathology. A higher GSI score indicates higher levels of psychological distress. Page 5, now lines 174-197.
- We have also added the following statement for the ethical considerations, as requested: “Participants provided verbal informed consent prior to their involvement in the study, and ethical approval was granted by the Ethics Committee of the University Hospital of Ioannina (protocol code 3263, dated 1st February 2019). Participants were also as-signed unique identifiers to anonymize their data, and only authorized researchers had access to the raw data”. Page 5, now lines 198-202.
Results are clearly presented, however, it is advisable to review the following aspects:
- In table 1, to clarify the meaning of "a", and to explain in line 169 how "internal reliability" was analyzed;
- We agree with the reviewer. In Table 1, the symbol 'α' refers to Cronbach's alpha coefficient, which is a widely recognized measure for evaluating the reliability of scales used in psychological assessments. Now we have added a note in the Table 1 as well we modified the first sentence as follows: “The psychometric scales used in this study demonstrated acceptable internal reliability, as evidenced by the examination of Cronbach's alpha (a) coefficient(Cronbach & Meehl, 1995)” Page 6, now lines 241-243. Additional notes added to Table 1.
- to revise the information presented in lines 201/202 and correspondence with figure 2.
- As per answer to Reviewer 1, we have now added that ‘Furthermore, our analysis showed some significant interaction effects (Table 2), which are further illustrated in Figure 2, Panels A, B, and C. A noteworthy interaction between critical cere experience and urban size of ICU location concerning neuroticism was observed, suggesting that ICU personnel in minor cities tend to report higher levels of neuroticism across all experience levels than in metropolitan cities, with the greatest difference seen in the group with more than 10 years of experience (Figure 2, Panel A). Additionally, the analysis revealed a significant interaction between critical care experience and profession regarding psychoticism and extraversion. Specifically, as experience in the ICU increases, nurses exhibited higher levels of psychoticism and lower levels of extraversion (Figure 2, Panels B and C).” Page 8, now lines 286-295). This revised text is now in line with Figure 2.
The conclusions are well supported by the data, and the authors discuss the limitations of the study.
- We thank the reviewer.
Overall, the manuscript publication is recommended.
- We thank the reviewer.
#in the attached file there is a cover letter and the complete response to all reviewers´comments

Round 2
Reviewer 2 Report
Comments and Suggestions for Authors
Dear Authors,
Thank you for making the needed revisions to improve this manuscript.
Best regards.
Minor editing of the English language is required.